# Clinical Presentation of a Patient with a Congenital Disorder of Glycosylation, Type IIs (*ATP6AP1*), and Liver Transplantation

**DOI:** 10.3390/ijms24087449

**Published:** 2023-04-18

**Authors:** Natalia Semenova, Olga Shatokhina, Olga Shchagina, Elena Kamenec, Andrey Marakhonov, Anna Degtyareva, Natalia Taran, Tatiana Strokova

**Affiliations:** 1Research Centre for Medical Genetics, 115522 Moscow, Russia; 2National Medical Research Center for Obstetrics, Gynecology and Perinatology Named after V.I. Kulakov, Ministry of Health of the Russian Federation, 127994 Moscow, Russia; 3Department of Neonatology, I.M. Sechenov First Moscow State Medical University, 119435 Moscow, Russia; 4Federal Research Center of Nutrition and Biotechnology, 109240 Moscow, Russia

**Keywords:** *ATP6AP1*, liver transplantation, cholestasis, X-linked primary immunodeficiency

## Abstract

The congenital disorder of glycosylation type IIs (ATP6AP1-CDG; OMIM# 300972) is a rare X-linked recessive complex syndrome characterized by liver dysfunction, recurrent bacterial infections, hypogammaglobulinemia, and defective glycosylation of serum proteins. Here, we examine the case of a 1-year-old male patient of Buryat origin, who presented with liver dysfunction. At the age of 3 months, he was hospitalized with jaundice and hepatosplenomegaly. Whole-exome sequencing identified the *ATP6AP1* gene missense variant NM_001183.6:c.938A>G (p.Tyr313Cys) in the hemizygous state, which was previously reported in a patient with immunodeficiency type 47. At the age of 10 months, the patient successfully underwent orthotopic liver transplantation. After the transplantation, the use of Tacrolimus entailed severe adverse effect (colitis with perforation). Replacing Tacrolimus with Everolimus led to improvement. Previously reported patients demonstrated abnormal N- and O-glycosylation, but these data were collected without any specific treatment. In contrast, in our patient, isoelectric focusing (IEF) of serum transferrin was performed only after the liver transplant and showed a normal IEF pattern. Thus, liver transplantation could be a curative option for patients with ATP6AP1-CDG.

## 1. Introduction

Congenital disorders of glycosylation (CDG) are a clinically and genetically heterogeneous group of disorders caused by errors in different steps along glycan modification pathways [1]. To date, more than 130 types of CDG have been identified, and more than 140 genes are associated with different types of CDG, which are characterized by a broad spectrum of clinical manifestations and severity [2]. CDG type IIs is caused by pathogenic variants of the *ATP6AP1* gene and characterized by a defect in N-glycosylation and, in some cases, a defect in O-glycosylation. The *ATP6AP1* gene encodes for the ATPase H+ transporting protein. This is an accessory subunit of the vacuolar V-ATPase proton pump complex that regulates pH homeostasis in cells [3]. Defects in certain subunits and accessory proteins of V-ATPase can cause congenital disorders of glycosylation [4]. In 2016, Jansen et al. described a novel ATP6AP1-linked immunodeficiency and identified disease-causing pathogenic variants in *ATP6AP1* in 11 male patients with abnormal protein glycosylation [5]. The common clinical symptoms displayed by this cohort of patients included immune abnormalities and hepatopathy. Recurrent bacterial infections were associated with hypogammaglobinemia. Hepatopathy varied from mild hypertransaminasemia to cirrhosis and end-stage liver failure. In addition, common laboratory abnormalities included leukopenia, slightly elevated serum transaminases, low serum copper and ceruloplasmin, and high alkaline phosphatase.

Here, we present the case of a 1-year-old male patient of Buryat origin with liver dysfunction that eventually led to liver transplantation.

## 2. Case Presentation

### 2.1. Clinical Data

The family of the affected male were clinically examined at the National Medical Research Center for Obstetrics, Gynecology, and Perinatology named after the academician V.I. Kulakov, as well as at the Federal Research Center of Nutrition and Biotechnology and the Research Centre for Medical Genetics, Moscow, Russia.

### 2.2. Genetic Testing

Blood samples from the proband and his unaffected parents were collected, and genomic DNA was extracted using the Wizard^®^ Genomic DNA Purification Kit (Promega, Madison, WI, USA) according to the manufacturer’s recommendations. Clinical exome sequencing was performed for the proband. Target enrichment with a SeqCap EZ HyperCap Workflow solution capture array (Roche Sequencing Solutions, Santa Clara, CA, USA), including the coding regions of 6640 genes currently described as clinically significant in the OMIM and the Human Gene Mutation Database (HGMD), and sequencing were carried out using Illumina NextSeq 500. The sequencing data were processed using a standard computer-based algorithm from Illumina and BaseSpace software (Enrichment 3.1.0). The sequenced fragments were visualized with Integrative Genomics Viewer (IGV) software (© 2023–2018 Broad Institute and the Regents of the University of California, Berkeley, CA, USA). Filtering of the variants was based on their frequency of less than 1% in gnomAD and coding region sequence effects such as missense, nonsense, coding indels, and splice sites. The variants’ clinical significance was evaluated according to the guidelines for massive parallel sequencing (MPS) data interpretation [6]. Automatic Sanger sequencing was carried out on ABIPrism 3500 (Applied Biosystems, Waltham, MA, USA) according to the manufacturer’s protocol. The primer sequences (ATP6AP1F CTCTAAGATGCCAAAGGCCCTC ATP6AP1R CTTCGTCTCTCAACCACTAGCC) were chosen according to the GeneBank database (NM_001183.6). The size of the PCR fragment was 491 base pairs.

### 2.3. Ethical Consideration

The study was approved by the local ethics committee of the Research Centre for Medical Genetics (the approval number 2018-1/3).

### 2.4. Clinical Evaluation

The proband was an affected 1-year-old boy born to non-consanguineous Buryat parents. The pregnancy was complicated by proteinuria in the 3rd trimester. The boy was born at 39 weeks of gestation. The birth weight was 3670 g (Z-score 1.65 SD), the birth length was 53 cm (Z-score 0.64 SD), and the Apgar score was 8/9 (Table 1). The perinatal period was normal. Myotonic syndrome was diagnosed at the age of one month. At the age of three months, the patient was admitted to the local hospital with jaundice and hepatosplenomegaly. Laboratory evaluation revealed thrombocytopenia, hypoglycemia, elevated transaminases, as well as a significant increase in alkaline phosphatase (more than tenfold) and in alpha-fetoprotein (Table 1). Based on the clinical picture and laboratory findings, the patient was diagnosed with idiopathic hepatitis (probably due to an inherited metabolic disease). Tandem mass spectrometry (MS/MS analysis) of acylcarnitines and amino acids in plasma detected elevated tyrosine levels, with a normal level of succinylacetone. Oncologists excluded hepatic tumors, using ultrasound and MRI (magnetic resonance imaging) methods of visualization. During hospitalization, the patient was treated with ursodeoxycholic acid at a dose of 25 mg/kg/day with clinical improvement.

One month later, the child was hospitalized due to a progression of hepatosplenomegaly and abnormal liver function tests. Laboratory investigations revealed mild anemia, secondary thrombocytopenia (probably due to hypersplenism and lack of thrombopoietin), cholestasis, and impaired coagulation.

At the age of 7 months, the child was first examined in the National Medical Research Center for Obstetrics, Gynecology, and Perinatology. On examination, jaundice was noted, and the liver was slightly increased, but it was dense. The spleen was significantly enlarged (+7 cm felt below the left costal margin). Diffuse muscle hypertonia was noted. The examination revealed cholestasis, elevated transaminases and alkaline phosphatase, as well as hypocoagulation (see Table 1). Ultrasonography demonstrated a mild hepatomegaly (the liver length in midclavicular line was 86 mm, +1.5 SD) and a significant enlargement of the spleen (the spleen length was 84 mm, +2.5 SD) without signs of portal hypertension. There were no esophageal varicose veins. Differential diagnosis was carried out between hereditary metabolic diseases, including lysosomal storage diseases, peroxisomal disorders, and others. A gas chromatography–mass spectrometry blood analysis showed normal concentrations of very-long-chain fatty acids and phytanic acid. An NGS-based panel analysis including 52 genes associated with cholestasis was also performed. There were no causative variants in the genes included in the panel.

At the age of 10 months, due to the development of liver failure, the child successfully underwent orthotopic living-donor liver transplantation. His father was the donor. The patient received Tacrolimus with monitoring of the whole-blood concentrations of the drug and dosage adjustments. On the 10th postoperative day, he developed an intestinal perforation of the terminal ileum, lying within 15 cm of the ileocecal valve. A right hemicolectomy was performed. It was regarded as an infectious complication due to immunosuppressive therapy and immunodeficiency.

Few days after the surgery, loose stools, abdominal distension, and bloating appeared. Over the next two months, the patient’s condition deteriorated rapidly, with a developing fever and cough. The patient was admitted to the hospital with acute respiratory distress syndrome, acute kidney injury, dyspeptic syndrome, and pancytopenia. On admission, his blood test results were as shown in Table 1. Due to drug intolerance and adverse side effects, the treatment with Tacrolimus was discontinued. The patient received a transfusion of fresh frozen plasma, as well as a correction of electrolyte disorders and hypoalbuminemia. Therapy with antibiotics and immunoglobulin was started, leading to clinical improvement.

One month later, at the age of 1 year 4 months, the patient’s weight was 8.6 kg (with a Z-score of 1.1 SD), and his length was 76.5 cm (with a Z-score of 0.77 SD) (Figure 1); the main symptoms were frequent stool and poor weight gain. A colonoscopy showed diffuse active colitis. A colonic biopsy revealed focal surface erosions, increased infiltration of lymphocytes, and eosinophils in the lamina propria. A gastroscopy demonstrated gastritis. Given the patient’s stable condition, restarting the treatment with Tacrolimus with monitoring of the whole-blood concentrations of the drug and dosage adjustments was recommended. Eleven days after the start of the treatment with Tacrolimus, the patient developed fever, dyspeptic syndrome, and face swelling. Laboratory studies revealed anemia (hemoglobin of 79 g/L), thrombocytopenia (platelet cell count of 57 × 103/μL), neutropenia, hypoalbuminemia, hypokalemia, and an elevated level of CRP (150 mg/L). Bone marrow aspiration showed no signs of hemophagocytosis. The decision was made to switch from Tacrolimus to Everolimus at a starting dose of 0.25 mg twice a day. The patient received antibiotic therapy and intravenous fluids to correct electrolyte disturbances and hypoalbuminemia, with improvement.

### 2.5. Genetic Analysis

Whole-exome sequencing was performed to find the causative gene defect. As a result, the missense variant NM_001183.6:c.938A>G (p.Tyr313Cys) in the hemizygous state was identified in the *ATP6AP1* gene. The described variant was previously reported in a patient with immunodeficiency type 47, characterized by liver dysfunction, recurrent bacterial infections, hypogammaglobulinemia, and defective glycosylation of serum proteins. The variant was not registered in the gnomAD database or the control group of Russian patients’ exomes. The p.Tyr313Cys variant is predicted by SIFT, PolyPhen2, Provean, and MutationTaster to have a deleterious effect on protein function. According to the ACMG-AMP criteria, the mentioned variant is of uncertain significance (PM2, PP3, PP5). Sanger sequencing validated this variant as hemizygous in the proband (III.2) and heterozygous in his unaffected mother (II.3). No mutation was identified in the patient’s healthy brother and other members of the family from the mother’s side, excluding the grandfather. We could not test him because he died many years ago.

## 3. Discussion

The *ATP6AP1* gene, localized on the X chromosome, encodes a subunit of the proton-transporting V-ATPase enzyme, which plays an important role in numerous physiological processes in the human body. Therefore, mutations in genes encoding subunits of the V-ATPase enzyme are associated with numerous pathological conditions, including renal tubular acidosis, deafness, osteoporosis, and cancer [7,8,9]. According to currently existing research literature data, pathogenic variants in the *ATP6AP1* gene lead to X-linked recessive complex primary immunodeficiency syndrome type 47 (OMIM#300972). The core symptoms of this condition reported by Jansen et al. (2016) in a cohort of 11 males with hemizygous missense variants in the *ATP6AP1* gene include defective glycosylation of serum proteins and liver dysfunction with neonatal jaundice and hepatosplenomegaly [5]. Six patients from three families also presented with neurological symptoms, including seizures, mild intellectual disability, and behavioral abnormalities. Dimitrov et al. (2018) and Witters et al. (2018) reported two patients with IMD47, both of whom had hepatopathy (elevated levels of transaminases and total bilirubin with a normal level of GGT; hyperechogenic liver parenchyma, normal liver size but enlarged size spleen (+3.7 SD) by ultrasonography), immune abnormalities, glycosylation defects, and cutis laxa [10,11]. Tvina et al. (2020) described a prenatal phenotype of an X-linked *ATP6AP1* gene mutation and the association of this gene mutation with increased nuchal translucency (NT), elevated alpha-fetoprotein (AF-AFP), positive acetylcholinesterase (AchE), and Aplasia Cutis Congenita [12]. In our case, in contrast, there were no pathological signs during the prenatal period.

In this case report, we described a 1-year-old male patient of Buryat ancestry with primary immunodeficiency and liver dysfunction that led to liver transplantation. Whole-exome sequencing identified the *ATP6AP1* gene missense variant NM_001183.6:c.938A>G (p.Tyr313Cys) in the hemizygous state, as previously reported by Jansen et al. in a patient with immunodeficiency type 47. Our patient had clinical symptoms and laboratory findings similar to the 4-year-old of Irish descent described by Jansen et al., including mild hepatomegaly and severe splenomegaly, as well as increased levels of transaminases. Interestingly, our patient did not have recurrent infections before the liver transplantation and immunosuppressive treatment. For the above-mentioned reasons, screening for N-glycosylation by isoelectric focusing of serum transferrin should be performed in patients with cholestasis, hepatomegaly, and severe splenomegaly without portal hypertension. The patient described by Jansen et al. demonstrated abnormal N- and O-glycosylation, while for our patient, isoelectric focusing of serum transferrin was performed only after the liver transplant and showed a normal IEF pattern. We suggest that a possible explanation for the normal glycosylation of transferrin in our patient might be a normalization of the glycosylation profile due to the liver transplant, as Mirian et al. reported when describing the first successful liver transplantation in a patient with a congenital disorder of glycosylation, after which, normal N-glycosylation of transferrin was found [13].

## 4. Conclusions

We discussed the case of a 1-year-old male patient of Buryat origin, who presented with liver dysfunction, caused by the *ATP6AP1* gene missense variant NM_001183.6:c.938A>G (p.Tyr313Cys) in the hemizygous state. At the age of 10 months, the child successfully underwent orthotopic liver transplantation. After the transplantation, the use of Tacrolimus entailed a severe side effect (colitis with perforation). A change in therapy from Tacrolimus to Everolimus led to improvement. This case report highlights the importance of performing biochemical screening and genetic tests in infants with hepatic dysfunction and cholestasis for differential diagnosis and successful therapy.

## Figures and Tables

**Figure 1 ijms-24-07449-f001:**
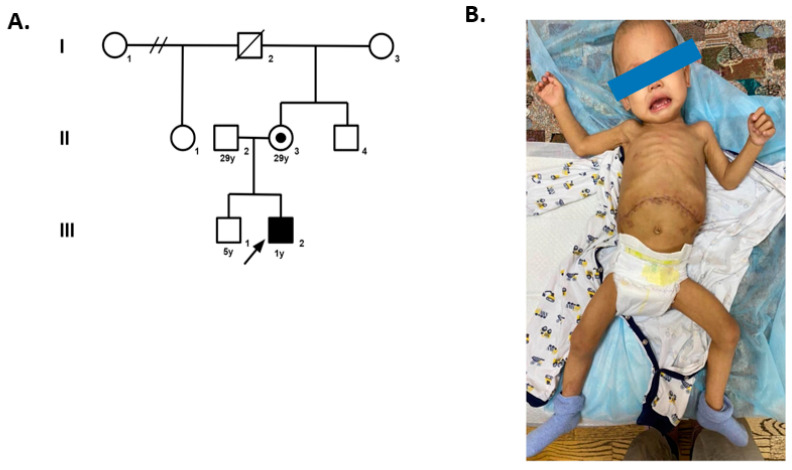
Pedigree chart of the patient (**A**) and patient’s general appearance at the age of 1 year (**B**).

**Table 1 ijms-24-07449-t001:** Anthropometric parameters and laboratory data (from birth to one year and four months).

	0 m	3 m	5 m	7 m	10 m	1 y 1 m	1 y 4 m	
Weight, g/SD	3670/0.64	7100/0.26	7800/1.44	7900/−0.06		8300/−1.58	8600/−1.1	
Length, cm/SD	53/1.65	62/0.25	69/1.44	72/2.01		74.5/−1.0	76.5/−0.77	
	Blood tests							
	No data	3 m	5 m	7 m	Liver transplantation	1 y 1 m	1 y 4 m	Normal ranges
hemoglobin		105	102	90		84	114	110–140 g/L
red blood cells		-	3.28	3.24		2.85	4.04	3.5–4.5 × 10^12^/L
white blood cells		-	6.6	5.8		6.9	5.28	6–17.5 × 10^9^/L
platelets		-	100	-		96	174	160–390 × 10^9^/L
total bilirubin		80	98.1	54.5		5.8	8.2	5–21 mkM/
direct bilirubin		23.8	75.6	24.1		2.9	4.5	<3.4 mkM/L
ALT		69	51	50.6		17.3	42	0–40 U/L
AST		190	183	133.5		33.9	34	0–40 U/L
ALP		4164	2523	1754		387	612	82–383 U/L
GGT		-	-	29.4		45.4	187	0–6 m: <204; 6–12 m: <34; 1–3 y:< 18 U/L
glucose		1.9	2.81	5.01		5.03	5.12	3.3–5.5 mM/L
urea		-	-	3.1		4.2	2.4	2.8–7.2 mM/L
cholesterol		-	5.1	2.26		1.86	1.86	3.2–5.2 mM/L
total protein		-	-	61.5		54.8	65.6	64–83 g/L
albumin		-	-	39.8		35.3	38.9	35–52 g/L
AFP		47,241	85,602	6306		-	-	0.5–50,000 IU/mL
fibrinogen		-		1.1		-	3.18	2–4 g/L
prothrombin index		58.1	32	38		85	-	81–138%
aPTT		75.6	100.2	51		27	44.3	25–35 s
INR		1.36	-	3.2		1.13	1.1	0.88–1.1

Note: m: month; y: years; ALT: alanine aminotransferase; AST: aspartate aminotransferase; ALP: alkaline phosphatase; GGT: gamma-glutamyltranspeptidase; AFP: alpha-fetoprotein; aPTT: activated partial thromboplastin time, INR: international normalized ratio.

## Data Availability

The datasets used and/or analyzed during this study are available from the corresponding author upon reasonable request.

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
