# Peer review of "Clinical Presentation of a Patient with a Congenital Disorder of Glycosylation, Type IIs (ATP6AP1), and Liver Transplantation"

_ijms, 2023, doi:10.3390/ijms24087449_

Round 1
Reviewer 1 Report
The mansucript of Natalia Semenova et al: seems to be interesting but there are many corrections to be made
Titre, correct: Clinical presentation of a patient with a congenital disorder of glycosylation, type IIs (ATP6AP1) and liver transplantation
Abstract
Delete this sentence in abstract: which was previously reported by Jansen et al. in a patient with immunodeficiency type 47.
Rephrase Line20-23: For our patient, isoelectric focusing (IEF) of serum transferrin was performed only after the liver transplant and showed a normal IEF pattern.
Keywords: liver failure?????
Introduction:
Start by congenital disorder of glycosylation IIs and then you talk about the gene ATP6AP1 by mentioning the link between the two.
Only one reference quote in the introduction: that's not much
Bring back this sentence in the results: Whole-exome sequencing identified ATP6AP1 gene missense variant NM_001183.6:c.938A>G (p.Tyr313Cys) in the hemizygous state.
The authors did not say anything about the liver transplantation.
Line 51-53, Put this in a separate point in 2.3. Ethical Consideration: The study was approved by the local ethics committee of the Research Centre for Medical Genetics (the approval number 2018-1/3).
DNA was extracted by standard methods:the extraction kit must be specified.
Line 56: complete the sentence "Clinical exome sequencing of ATP6AP1 was performed for the proband.
MM is very poor in information. It tells you a bit about how the WHole exome was made. What bioinformatics tools are used to detect mutations.
Sanger sequencing is not well described: how you confirmed the variants? you designed primers to target specific parts? What size base pairs were sequenced?
Results:
MS/MS analysis, MRI methods: abreaviation , attention
Line 130: Table 1. Clinical and laboratory data: statistical tests of p-value should be calculated to see if there is a difference between these figures in the same line
Line 149-160: this has already been said in the introduction. : delete
In the discussion, you just have to put forward your results and compare them with previous studies: that's all!
Author Response
First of all, we eager to thank you for detailed and thorough analysis of our manuscript as well as valuable comments and recommendations to improve it. Thereunder you could find responses to each comment.
Titre, correct: Clinical presentation of a patient with a congenital disorder of glycosylation, type IIs (ATP6AP1) and liver transplantation
Answer 1: The title is corrected accordingly.
Abstract
Delete this sentence in abstract: which was previously reported by Jansen et al. in a patient with immunodeficiency type 47.
Answer 2: We have deleted the citation from the abstract but we think that information of incomplete similarity of previously reported patient is important to mention it in the Abstract.
Rephrase Line20-23: For our patient, isoelectric focusing (IEF) of serum transferrin was performed only after the liver transplant and showed a normal IEF pattern.
Answer 3: We have rephrased this sentence and have added some additional in formation to make the conclusion clearer.
Keywords: liver failure?????
Answer 4: We have changed this keyword.
Introduction:
Start by congenital disorder of glycosylation IIs and then you talk about the gene ATP6AP1 by mentioning the link between the two.
Answer 5: We have changed the text accordingly.
Only one reference quote in the introduction: that's not much
Answer 6: We have added references to the Introduction.
Bring back this sentence in the results: Whole-exome sequencing identified ATP6AP1 gene missense variant NM_001183.6:c.938A>G (p.Tyr313Cys) in the hemizygous state.
Answer 7: We have changed the text accordingly.
The authors did not say anything about the liver transplantation.
Answer 8: Liver transplantation is one of the central parts of our manuscript. There is a separate paragraph (lines 98-104) with the description of this treatment.
Line 51-53, Put this in a separate point in 2.3. Ethical Consideration: The study was approved by the local ethics committee of the Research Centre for Medical Genetics (the approval number 2018-1/3).
Answer 9: We have changed the text accordingly.
DNA was extracted by standard methods:the extraction kit must be specified.
Answer 10: We have changed the text accordingly.
Line 56: complete the sentence "Clinical exome sequencing of ATP6AP1 was performed for the proband.
MM is very poor in information. It tells you a bit about how the WHole exome was made. What bioinformatics tools are used to detect mutations.
Answer 10: We have significantly extended the M&Ms section according to the request.
Sanger sequencing is not well described: how you confirmed the variants? you designed primers to target specific parts? What size base pairs were sequenced?
Answer 11: We have added the information about PCR primers used for Sanger validation of the variant in the proband.
Results:
MS/MS analysis, MRI methods: abreaviation , attention
Answer 12: We have deciphered these abbreviations.
Line 130: Table 1. Clinical and laboratory data: statistical tests of p-value should be calculated to see if there is a difference between these figures in the same line
Answer 13: There was no aim to perform any statistical analysis on these isolated parameters in in a single patient. The data presented was shown only to demonstrate dynamics of clinical symptoms.
Line 149-160: this has already been said in the introduction. : delete
Answer 14: We have deleted repeated fragments from introduction, thus there is no any redundancy in text in Results section.
In the discussion, you just have to put forward your results and compare them with previous studies: that's all!
Answer 15: We have deleted the redundant information which is repeated in Introduction section. In the Discussion we left only information about clinical spectrum associated with the ATP6AP1 gene pathogenic variants and compare clinical picture of our patient with those previously reported. We also have made some suggestions about liver transplantation influence on the clinical outcome of patients with ATP6AP1 – CGD and made it important to perform selective screening for demonstrated abnormal N- and O-glycosylation in patients with cholestasis and severe splenomegaly.
Reviewer 2 Report
Semenova et al. reported that clinical presentation of a patient with a congenital disorder of glycosylation IIs and liver transplantation. This report seems important in this area.
1. Authors should make figure 1 legend.
Author Response
First of all, we eager to thank you for detailed and thorough analysis of our manuscript as well as valuable comments and recommendations to improve it. Thereunder you could find responses to each comment.
- Authors should make figure 1 legend.
Answer 1: Thank you very much for your attention. We have also noticed absent letters indicators on the Figure 1 which we have also corrected.
Reviewer 3 Report
The manuscript presents an interesting case of inborn errors of metabolism affecting the glycosylation process. I recommend several improvements:
Row 69- replace 1 month with one month.
Row 85 – not necessary to repeat ‘named after Academician….”
Row 86- is it possible to give details regarding the degree - size for hepatomegaly?
Row 89- please write after hypo coagulation: (see Table 1).
Insert here (after row 89 the Table 1 – being the first time when is referred
Row 90: please give the dimension of the spleen.
Row 107: after word ‘surgery’ ..should write (how long after surgery - how many months??)..
Row 115: .please insert what is appropriate in the end of the sentence .leading to improvement (clinical and/or biochemical signs improvement?)
For the Table 1: the title that is more appropriate: Anthropometric parameters and laboratory data (from birth to one year and four months)
In the Table 1:
- please include a column with normal values of the laboratory that did the measurements
- write in the first row, first column: Age, and then, replace m with month, y with years (or put m and y in the legend of the table)
- include an arrow writing ‘Liver transplantation’ between column for 7 months and 1 year 1 month.
Row 161: delete a space before “the core symptoms…”
Row 167: please specify which type of hepatopathy..
Row 169: please define NT (as well in the end of the manuscript where is the list of abbreviations)
Row 188: please specify which type of glycosylation defect are the authors referring to (what is in the reference 10).
Row 195:..as is mentioned in another part of the manuscript ..Tacrolimus with Everolimus and antibiotics led to improvement.
Row 196. To complete: ..performing biochemical screening and molecular tests (if possible) in infants..
Author Response
First of all, we eager to thank you for detailed and thorough analysis of our manuscript as well as valuable comments and recommendations to improve it. Thereunder you could find responses to each comment.
Row 69- replace 1 month with one month.
Answer 1: Thank you very much for your attention. We have corrected the text accordingly.
Row 85 – not necessary to repeat ‘named after Academician….”
Answer 2: We have corrected the text accordingly.
Row 86- is it possible to give details regarding the degree - size for hepatomegaly?
Answer 3: We have added liver size by ultrasonography in the text.
Row 89- please write after hypo coagulation: (see Table 1).
Answer 4: We have corrected the text accordingly.
Insert here (after row 89 the Table 1 – being the first time when is referred
Answer 5: We have corrected the text accordingly.
Row 90: please give the dimension of the spleen.
Answer 6: We have corrected the text accordingly.
Row 107: after word ‘surgery’ ..should write (how long after surgery - how many months??)..
Answer 7: We have specified time period.
Row 115: .please insert what is appropriate in the end of the sentence .leading to improvement (clinical and/or biochemical signs improvement?)
Answer 7: We have specified the improvement was clinically noticed.
For the Table 1: the title that is more appropriate: Anthropometric parameters and laboratory data (from birth to one year and four months)
Answer 6: We have corrected the title of the Table 1 accordingly.
In the Table 1:
- please include a column with normal values of the laboratory that did the measurements
Answer 7: We have added the normal values to the Table 1.
- write in the first row, first column: Age, and then, replace m with month, y with years (or put m and y in the legend of the table)
Answer 8: We have added this information to the note to the Table 1.
- include an arrow writing ‘Liver transplantation’ between column for 7 months and 1 year 1 month.
Answer 9: We have added a separate column with the age of the surgery to the Table 1.
Row 161: delete a space before “the core symptoms…”
Answer 9: Corrected.
Row 167: please specify which type of hepatopathy.
Answer 10: We have specified the type of hepatopathy.
Row 169: please define NT (as well in the end of the manuscript where is the list of abbreviations)
Answer 11: We have added this information within the text as well as into the abbreviations list.
Row 188: please specify which type of glycosylation defect are the authors referring to (what is in the reference 10).
Answer 12: We have specified this information.
Row 195:..as is mentioned in another part of the manuscript ..Tacrolimus with Everolimus and antibiotics led to improvement.
Answer 13: The main point of the manuscript is that only change in therapy from Tacrolimus to Everolimus led to improvement of clinical symptoms. Antibiotics were prescribed before this change without improvement as well as during this change and played only supportive role. We strongly believe that the improvement is a result of change in immunosuppressive therapy.
Row 196. To complete: ..performing biochemical screening and molecular tests (if possible) in infants..
Answer 14: Thank you very much for your attention. We have corrected the text accordingly.